# Self-Rated Health as a Predictor of Mortality in Older Adults: A Systematic Review

**DOI:** 10.3390/ijerph20053813

**Published:** 2023-02-21

**Authors:** Moustapha Dramé, Eléonore Cantegrit, Lidvine Godaert

**Affiliations:** 1EpiCliV Research Unit, Medical School, University of the French West Indies, 97261 Fort-de-France, France; 2Department of Clinical Research and Innovation, University Hospitals of Martinique, 97261 Fort-de-France, France; 3Department of Geriatrics, General Hospital of Valenciennes, 59300 Valenciennes, France

**Keywords:** self-rated health, mortality, older adults, prediction

## Abstract

The aim of this study was to investigate the link between self-reported health (SRH) and mortality in older adults. In total, 505 studies were found in PubMed and Scopus, of which 26 were included in this review. In total, 6 of the 26 studies included did not find any evidence of an association between SRH and mortality. Of the 21 studies that included community dwellers, 16 found a significant relationship between SRH and mortality. In total, 17 studies involved patients with no specific medical conditions; among these, 12 found a significant link between SRH and mortality. Among the studies in adults with specific medical conditions, eight showed a significant association between SRH and mortality. Among the 20 studies that definitely included people younger than 80 years, 14 found a significant association between SRH and mortality. Of the twenty-six studies, four examined short-term mortality; seven, medium-term mortality; and eighteen, long-term mortality. Among these, a significant association between SRH and mortality was found in 3, 7, and 12 studies, respectively. This study supports the existence of a significant relation between SRH and mortality. A better understanding of the components of SRH might help guide preventive health policies aimed at delaying mortality in the long term.

## 1. Introduction

Numerous studies have investigated the predictive value of self-reported health (SRH) on mortality or adverse health outcomes in both young and old adults [1,2,3]. Overall, the results of these studies, particularly regarding the link between SRH and mortality, widely vary according to the age and sex of the population studied, the length of follow-up, or the presence or absence of specific diseases [4]. It is, therefore, difficult to know with any certainty what weight should be given to patients’ SRH. This difficulty is particularly marked among older adults, who are often frail and multimorbid, and who may have a life expectancy that is limited by one or more chronic diseases. SRH is a valuable assessment, because it covers multiple components and is easy to collect. Several authors [5,6] have shown the multiple domains are encompassed by the term self-reported health. However, the contribution of each individual component to the overall evaluation remains to be determined and seems to vary according to the context (gender, socio-economic or educational level, age category, religion, etc.). The evaluation of SRH yields a more comprehensive view of an individual’s health and may be more accurate than a purely medical evaluation. Moreover, it allows physicians to understand complex predictive factors of health, such as chronic inflammatory status [7,8]. Finally, SRH can be evaluated by asking a single, simple question [9].

In this systematic review, we aimed to determine whether there is a significant link between SRH and mortality in older adults.

## 2. Methods

### 2.1. Search Strategy

Before launching the literature search, we ensured that no systematic review had previously been conducted on this specific topic and in this particular population, by means of verification in PubMed, Scopus, Prospero, and the Cochrane library.

This was only a systematic review. A comprehensive literature search was performed in PubMed and Scopus. The search covered all publications up to and including 23 March 2022, with no specific start date specified. Search terms were defined by two senior researchers (L.G. and M.D.) and included the following keywords in the title and/or the abstract: (“obesity paradox” OR “reverse epidemiology” OR “body mass index”) AND (mortality OR death OR survival) (“self-rated health” OR “perceived health” OR “subjective health” OR “health report” OR “quality of life”) AND (mortality OR outcome OR survival OR death) AND (Age OR old OR elder*). Filters were applied to select studies in the English or French language and studies only including human subjects and to exclude the following publication types: reviews, case reports and case series, editorials, and correspondence. Reference lists were manually checked for additional studies. Study selection was performed in accordance with the Preferred Reporting Items for Systematic Reviews and Meta-Analyses (PRISMA) guidelines. This study was registered with PROSPERO (an International prospective register of systematic reviews), under the number CRD42022329082.

### 2.2. Study Selection Criteria

Study eligibility criteria were defined prior to performing the literature search by two senior researchers (L.G. and M.D.) according to the PICOS framework. Studies were eligible for inclusion if they reported data on self-rated health. The population of the studies included people aged 65 years or older, of any sex, ethnicity, or living place. The groups to be compared were defined according to their levels of self-rated health (SRH). The outcome was death, whatever the timepoint. Basic science articles, reviews, case reports and case series, editorials, and correspondence were excluded.

### 2.3. Data Extraction

Data analysis was performed using Covidence systematic review software© (Veritas Health Innovation, Melbourne, Australia), available at www.covidence.org (accessed on 23 March 2022). After eliminating duplicates, two senior researchers (L.G. and M.D.) independently reviewed the titles and abstracts of all articles. In case of disagreement about whether or not to include an article, the case was discussed until consensus was reached. Overlap between studies in the results reported was checked. We independently extracted the data, using the same data extraction form. For descriptive analyses, the following data were extracted: publication year, country where the study was conducted, study design, study setting, medical condition (if any), sample size, and age (mean or median and their statistical dispersion parameters, when available). To analyse the relation between SRH and mortality, the following information was collected: outcome (death or survival), type of analysis (whether adjusted or not), SRH levels, statistical estimates (hazard ratios, odds ratios, rate ratios, and rates) and their respective 95% confidence intervals (95% CIs), and the level of significance (*p*-values).

### 2.4. Quality Assessment

The quality of the included studies was assessed independently by two researchers (L.G. and M.D.) using the Newcastle–Ottawa Scale (NOS) [10]. The NOS consists of three quality parameters: selection, comparability, and outcome assessment. The “selection” criterion is scored between 0 and 4 points; the “comparability” criterion is scored between 0 and 2 points; and the “outcome” criterion is scored between 0 and 3 points. The sum of the scores of these three criteria gives an NOS total score between 0 and 9 points. NOS scores of 7 or over were considered to be of high quality, while 5–6 indicated moderate quality, and scores under 5 indicated low quality. Disagreement was resolved by means of a joint review of the manuscript to reach consensus, and the opinion of a third researcher was requested when necessary. When appropriate and possible, certain parameters were calculated from available data (e.g., pooled mean age and/or standard deviations, odds ratios, rate ratios, etc.).

## 3. Results

In total, 505 studies were identified during the literature search (Figure 1). Among these, 195 duplicates were excluded. After examination of the titles and abstracts of the remaining 310 studies, 98 articles were retained for full-text assessment. After reading the full text of these 98 studies, 72 were excluded for one or more of the following reasons: inappropriate age of the study population, wrong study design, or wrong outcome. Thus, 26 studies were included in the final review.

Table 1 summarizes the characteristics of the studies included in the review. All studies were observational cohorts. The average age of the population included in the studies was >80 years in two articles [11,12] and was not specified in five articles [13,14,15,16,17]. The two articles [15,16] with a mean population age of over 80 were performed on the same cohort, with evaluation of mortality at different timepoints.

The main results of the included studies are summarized in Table 2. As shown in Table 2, 6 of the 26 studies did not find any evidence of an association between SRH and mortality [2,25,27,29,32,33]. Among the 21 studies that included community dwellers, 16 found a significant relationship between worse SRH and higher mortality rates [13,14,16,17,18,19,20,21,22,23,24,26,28,30,31,35]. A total of 17 studies involved patients with no specific medical conditions; among them, 12 found a significant link between worse SRH and higher mortality rates [13,14,15,16,17,18,19,20,21,22,24,30]. Among the studies including individuals with specific medical conditions, eight showed a significant association between SRH and mortality [11,12,23,26,28,31,34,35]. When only specific mortality was considered (six studies), the relationship with SRH was always significant [13,15,16,17,23,26]. Two studies involved people over the age of 80 years. They both showed a significant association between SRH and mortality [11,12]. Among the 20 studies that definitely included people younger than 80 years (but older than 65), 14 found a significant association between SRH and mortality [17,18,19,20,21,22,23,24,26,28,30,31,34,35]. Of the 26 studies, 4 examined short-term mortality (<one year), while 7 examined medium-term mortality (one to five years), and 18 studied long-term mortality (five years or over). Of these, a significant association between SRH and mortality was found in 3 [11,12,20], 7 [11,16,19,20,28,34,35], and 12 studies [13,14,15,17,18,21,22,23,24,26,30,31], respectively. When SRH was considered as a dichotomous variable (in 11 studies), it was significantly associated with mortality in 9 cases [11,12,18,19,23,24,28,30,31], and when considered as a non-dichotomous variable (in 17 studies), the association between worse SRH and higher mortality rates was significant in 12 cases [13,14,15,16,17,18,20,21,22,26,34,35].

The quality of the included studies, as assessed using the NOS, is summarized in Table 3. The quality was considered high for all 26 studies.

## 4. Discussion

In this systematic review of the predictive relationship between self-rated health (SRH) and mortality in people aged 65 years or over, we included 26 studies [2,11,12,13,14,15,16,17,18,19,20,21,22,23,24,25,26,27,28,29,30,31,32,33,34,35], of which 4 investigated short-term mortality (≤1 year) [2,11,12,20], 8 investigated medium-term (>1 year and <5 years) mortality [11,16,19,20,28,32,34,35], and 18 investigated long-term mortality (≥5 years) [13,14,15,17,18,19,21,22,23,24,25,26,27,29,30,31,32,33]. Five articles studied the relationship between SRH and mortality at several timepoints [11,18,19,20,32]. The majority of articles concerned populations without a specific medical condition at inclusion [13,14,15,16,17,18,19,20,21,22,24,25,27,29,30,32,33].

For the two studies that included people aged 80 years or over, the authors showed a significant relationship between SRH and all-cause mortality at each timepoint (6 weeks; 6 months; and 1, 2 and 3 years). However, it seems difficult to extrapolate these results, as they all concern the same population, hospitalised via the emergency department for an acute condition.

In the community-dwelling population, with a mean age of <80 years at inclusion, not selected for the presence of any specific pathology, our systematic review supports a predictive relationship between SRH and all-cause mortality at each time point. Of 17 articles [13,14,15,16,17,18,19,20,21,22,24,25,27,29,30,32,33] studying all-cause mortality, 12 [13,14,15,16,17,18,19,20,21,22,24,30] found a predictive relationship between SRH and death (i.e., 70.5% of the articles studied). Three articles [16,19,20] were specific to medium-term and nine [13,14,15,17,18,21,22,24,30] to long-term all-cause mortality. These results are consistent with previous studies of younger adults showing that SRH is predictive of all-cause mortality in the medium (<5 years) [36,37] and long term [3]. The persistence of a long-term predictive link is remarkable in the elderly population, as these are often fragile individuals, with multiple causes of death.

SRH is a composite concept that encompasses medical, social, cultural, religious, ethnical, and individual dimensions. Several authors have attempted to characterise the different dimensions of health under the term “SRH” [1,5,6,38,39,40]. The share of each dimension in the overall subjective feeling varies from one individual to another, explaining the variable strength of the link between SRH and mortality according to gender, culture, ethnicity, socio-economic level. and even age group [33,36,41,42,43]. Zajacova et al. [44] showed that the individual criteria taken into account when assessing SRH varied from one sex to another as well as according to the period of life. Younger women tended to assess their SRH more unfavourably than men of the same age, while older women had a more favourable view of their SRH than men of the same age. This trend is even more salient if socio-economic factors (such as education, marital status, or income) are taken into account. As people age, the SRH is generally poorer, in both sexes, and this worsens as health problems and loss of autonomy increase. This illustrates the likely important role of medical criteria and functional status in the assessment of SRH with advancing age. Zajacova et al. [44] pointed out that all health indicators (physical health such as functioning or pain, mental health such as depressive symptoms, and health behaviours) are significantly associated with SRH, regardless of age or sex. Cott et al. [45] made the same observation in adult populations with one or more chronic diseases.

SRH is also associated with other factors known to predict outcome in the elderly population, such as interkeukin-6 (IL-6) [7]. Arnberg et al. [7] found that good or very good SRH was associated with low levels of systemic markers of inflammation in a population with a median age of 74 years (range of 60–93 at inclusion). Christian et al. [8] reported similar findings. Taken together, these data confirm that the collection of SRH in routine practice would be a simple and effective way of complementing the usual medical assessment to extrapolate an individual’s life trajectory.

Throughout life, including in the older population, the SRH seems to be a fairly accurate assessment of an individual’s functional capacities and even functional reserves for coping with the hazards of life, as evidenced by the predictive link with all-cause mortality demonstrated at all ages of adult life and at all timepoints. SRH is easily collected [9], even in people with mild-to-moderate cognitive impairment [31,46].

The methods used to collect SRH are variable. In our systematic review, some authors chose to assess the SRH on a value scale from excellent to very poor (excellent, very good, good, fair, poor, and very poor). Others chose to class SRH on a binary scale (SRH (excellent, very good or good) versus (fair, poor or very poor)). Of the 11 authors who evaluated SRH on a binary scale, 9 (i.e., 81.8%) found a predictive link between SRH and mortality in the short, medium, or long term [11,12,18,19,23,24,28,30,31]. Among the authors who treated the SRH according to a multiple choice scale, 12 (i.e., 70.6%) [13,14,15,16,17,18,20,21,22,26,34,35] showed a predictive link between SRH and mortality for at least one time point. The predictive capacity of the SRH with respect to mortality seems to be better when SRH is treated as a binary variable, most likely because there is greater statistical power with a dichotomous variable than with a non-binary, categorical one.

The predictive link between SRH and specific mortality in specific medical conditions seems to be more difficult to establish, because it is less well documented. In this systematic review, three articles investigated mortality linked to cancer [13,15,16], and two of them found a significant predictive relationship between SRH and cancer-related death in the medium [16] and long term [13]. Five articles investigated cardiovascular mortality [13,15,16,23,26], of which four [13,15,23,26] found that SRH significantly predicted cardiovascular death in the long term in a population with a mean age of <80 years.

## 5. Conclusions

SRH seems to be a good criterion for assessing the risk of mortality in the short, medium, or long term in a population of elderly subjects living at home according to the articles studied in this systematic review. SRH assessment is complementary to so-called objective medical measures. SRH is simple to collect, which makes it easy to use for health professionals and acceptable to the population. Its composite nature makes it possible to take into account an individual’s health in a global manner.

A better understanding of the components of SRH and their respective weight at each age might help to guide preventive health policies aimed at delaying mortality in the long term. However, there are currently no studies that have established that improving the criteria comprising SRH would make it possible to reduce mortality.

Moreover, as the weight of each criterion seems to vary according to the individual and the age considered, targeted interventions may not be very effective. The composite nature of the SRH concept should encourage us to implement comprehensive prevention strategies from the outset, individualised and variable over time for greater effectiveness.

Prevention strategies should be implemented early in the life of the individual and continue throughout life. The identification of poor SRH in a patient should prompt healthcare providers to promptly look for associated modifiable factors in an attempt to improve them.

## Figures and Tables

**Figure 1 ijerph-20-03813-f001:**
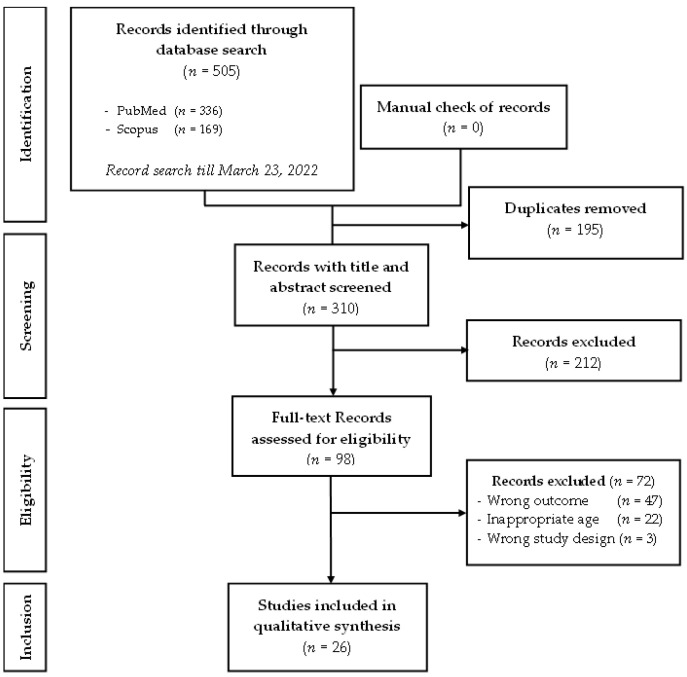
PRISMA flow diagram of the records included in the systematic review.

**Table 1 ijerph-20-03813-t001:** Description of the studies included in the present systematic review.

Author(s), Year	Country	Study Setting	Medical Conditions	Sample Size	Age (Years)
Wuorela et al., 2020 [18]	Finland	Community	No specific conditions (men only)	1008	70.0 ± 0.0 *
Godaert et al., 2018 [11]	France	Hospital, emergency	Hospitalised for an acute condition	223	85.1 ± 5.5 *
Godard-Sebillotte et al., 2016 [12]	France	Hospital, emergency	Hospitalised for an acute condition	223	85.1 ± 5.5 *
Mavaddat et al., 2016 [19]	England and Wales	Community including care homes	No specific conditions	11,957	74.8 ± 6.6 *
Brown et al., 2015 [20]	USA	Community	No specific conditions; exclusion of end-stage renal disease at baseline	191,001	75.0 ± x.x *
Gurland et al., 2014 [21]	USA	Community	No specific conditions	2128	76.0 ± 5.8 *
Shen et al., 2014 [15]	Hong Kong	Health centres	No specific conditions	66,814	≥65
Fernández-Ruiz et al., 2013 [22]	Spain	Community	No specific conditions	4958	74.1 ± 6.8 *
Puts et al., 2013 [2]	Canada	Hospital	Cancer	112	74.2 ± 6.0 *
Ernstein et al., 2011 [23]	Norway	Community	No specific conditions; exclusion of cardiovascular disease at baseline	5808	76.0 ± 4.9 *
Khang et al., 2010 [24]	Korea	Community	No specific conditions	1448	≥65
Ford et al., 2008 [25]	Australia	Community	No specific conditions (women only)	12,422	70–75 ^♦^
Johansson et al., 2008 [26]	Sweden	Community	Presence of signs or symptoms associated with chronic heart failure	448	73.0 ± 5.6 *
Okamoto et al., 2008 [27]	Japan	Community	No specific conditions	784	≥65
Lee et al., 2007 [28]	USA	Community	No specific conditions	6298	≥70
Van den Brink et al., 2005 [29]	Finland, Italy, and Netherlands	Community	No specific conditions (men only)	1141	76.5 ± 4.4 *
Baron-Epel et al., 2004 [30]	Israel	Community	No specific conditions	1138	77.5 (70–101) ^#^
Walker et al., 2004 [31]	Canada	Community	No specific conditions	8697	75.7 ± 7.1 *
Bath, 2003 [32]	UK	Community	No specific conditions	1042	≥65
Helmer et al., 1999 [33]	France	Community	No specific conditions	3660	75.2 (65–101) ^#^
Yu et al., 1998 [17]	Shanghai	Community	No specific conditions	3094	≥65
Leung et al., 1997 [34]	Taiwan	Long-term facility	No specific conditions	411	77.5 ± x.x *
Schoenfeld et al., 1994 [35]	USA	Community	Aging successfully	1037	70–79 ^♦^
Tsuji et al., 1994 [16]	Japan	Community	No specific conditions	2252	65–113 ^♦^
Pijls et al., 1993 [13]	Netherlands	Community	No specific conditions (men only)	783	65–85 ^♦^
Rakowski et al., 1993 [14]	USA	Community	No specific conditions	5630	≥70

* mean ± standard deviation; ^#^ mean (range); ^♦^ range; x: not defined.

**Table 2 ijerph-20-03813-t002:** Outcome and results of association between SRH and mortality in aged adults.

Author(s), Year	Outcome	Medical Conditions	Analysis	Results		
				SRH Levels	Estimates (95% CI)	*p*
Wuorela et al., 2020 [18]	5-year mortality	No specific conditions	aHR	Good/rather good	Reference	
				Poor	2.17 (1.42–3.31)	<0.001
	10-year mortality			Good	Reference	
				Rather good	2.29 (1.24–4.23)	0.009
				Poor	4.08 (2.14–7.77)	<0.001
	27-year mortality			Good	Reference	
				Rather good	1.19 (0.94–1.51)	0.14
				Poor	1.62 (1.23–2.13	<0.001
Godaert et al., 2018 [11]	6-month mortality	Hospitalised for an acute condition	aHR	Very good/good	Reference	
				Medium to very poor	2.7 (1.6–4.7)	0.0003
	1-year mortality			Very good/good	Reference	
				Medium to very poor	2.4 (1.5–4.0)	0.0006
	2-year-mortality			Very good/good	Reference	
				Medium to very poor	1.9 (1.3–2.9)	0.002
	3-year mortality			Very good/good	Reference	
				Medium to very poor	1.6 (1.1–2.4)	0.01
Godard-Sebillotte et al., 2016 [12]	6-week mortality	Hospitalised for an acute condition	aHR	Very good/good	Reference	
				Medium to very poor	2.61 (1.18–5.77)	0.02
Mavaddat et al., 2016 [19]	2-year mortality	No specific conditions, no prior history of stroke	aOR	Excellent/good	Reference	
				Fair/poor	1.7 (1.4–2.0)	S
		No specific conditions, prior history of stroke		Excellent/good	Reference	
				Fair/poor	1.1 (0.7–1.8)	NS
	13-year mortality	No specific conditions, no prior history of stroke	aHR	Excellent	Reference	
				Good	1.2 (1.0–1.4)	S
				Fair	1.3 (1.1–1.6)	S
				Poor	1.2 (0.9–1.7)	NS
		No specific conditions, prior history of stroke		Excellent	Reference	
				Good	0.8 (0.5–1.3)	NS
				Fair	0.8 (0.4–1.3)	NS
				Poor	1.1 (0.6–2.1)	NS
Brown et al., 2015 [20]	90-day mortality	No specific conditions, no end-stage renal disease	aHR	Excellent	Reference	
				Very good	1.00 (0.56–1.78)	NS
				Good	1.65 (0.95–2.85)	NS
				Fair	3.03 (1.73–5.30)	<0.001
				Poor	7.36 (4.08–13.25)	<0.001
	Maximum follow-up mortality (>2.5 years)		aHR	Excellent	Reference	
				Very good	1.13 (1.01–1.27)	<0.05
				Good	1.60 (1.43–1.79)	<0.001
				Fair	2.52 (2.25–2.83)	<0.001
				Poor	4.24 (3.73–4.82)	<0.001
Gurland et al., 2014 [21]	16-year survival	No specific conditions	aHR	Poor	Reference	
				Excellent	0.69 (0.54–0.89)	S
				Good	0.79 (0.63–0.99)	S
				Fair	0.77 (0.62–0.96)	S
Shen et al., 2014 [15]	10-year all-cause mortality	No specific conditions	aHR	Better	Reference	
				Normal	0.86 (0.81–0.91)	S
				Worse	0.91 (0.86–0.96)	S
	10-year cardiovascular disease mortality			Better	Reference	
				Normal	0.85 (0.77–0.94)	S
				Worse	0.84 (0.76–0.94)	S
	10-year stroke mortality			Better	Reference	
				Normal	0.83 (0.70–0.99)	S
				Worse	0.88 (0.76–1.05)	NS
	10-year ischemic heart disease mortality			Better	Reference	
				Normal	0.88 (0.74–1.03)	NS
				Worse	0.84 (0.71–0.99)	S
	10-year all-cancer mortality			Better	Reference	
				Normal	0.90 (0.81–0.99)	S
				Worse	0.97 (0.87–1.08)	NS
	10-year all-respiratory disease mortality			Better	Reference	
				Normal	0.85 (0.75–0.96)	S
				Worse	0.93 (0.82–1.06)	NS
Fernández-Ruiz et al., 2013 [22]	13-year all-cause mortality	No specific conditions	aHR	Very good	Reference	
				Good	0.95 (0.81–1.12)	NS
				Fair	1.22 (1.03–1.44)	<0.05
				Poor/very poor	1.39 (1.15–1.69)	<0.01
Puts et al., 2013 [2]	12-month mortality	Newly diagnosed cancer	aHR	Good/excellent	Reference	
				Fair/poor/very poor	1.33 (0.50–3.53)	NS
Ernstein et al., 2011 [23]	10-year IHD mortality	No specific conditions; exclusion of cardiovascular disease at baseline (men)	aHR	Very good/good	Reference	
				Fair/poor	1.23 (0.91–1.67)	NS
		No specific conditions; exclusion of cardiovascular disease at baseline (women)		Very good/good	Reference	
				Fair/poor	1.61 (1.14–2.29)	S
	10-year all-cause mortality	No specific conditions; exclusion of cardiovascular disease at baseline (men)		Very good/good	Reference	
				Fair/poor	1.42 (1.25–1.61)	S
		No specific conditions; exclusion of cardiovascular disease at baseline(women)		Very good/good	Reference	
				Fair/poor	1.60 (1.39–1.84)	S
Khang et al., 2010 [24]	Long-term mortality	No specific conditions, non-institutionalized population, men	aHR	Very good/good/fair	Reference	
				Very poor/poor	2.21 (1.47–3.33)	S
		No specific conditions, non-institutionalized population (women)		Very good/good/fair	Reference	
				Very poor/poor	2.05 (1.33–3.15)	S
Ford et al., 2008 [25]	Long-term mortality	No specific conditions (women only)	aHR	Excellent	Reference	
				Very good	1.04 (0.77–1.41)	NS
				Good	1.27 (0.95–1.70)	NS
				Fair	2.10 (1.56–2.83)	S
				Poor	3.83 (2.73–5.38)	S
Johansson et al., 2008 [26]	10-year cardiovascular mortality	Presence of signs or symptoms associated with chronic heart failure	aHR	Very good	Reference	
				Good	3.4 (1.4–7.8)	0.005
				Poor	4.1 (1.8–9.4)	0.001
Okamoto et al., 2008 [27]	6-year mortality	No specific conditions (men)	aHR	Fair/Poor	Reference	0.04 ^#^
				Good	0.63 (0.32–0.98)	
				Excellent	0.48 (0.14–1.07)	
		No specific conditions (women)		Fair/poor	Reference	0.40 ^#^
				Good	0.78 (0.41–1.33)	
				Excellent	0.74 (0.21–1.32)	
Lee et al., 2007 [28]	4-year mortality	No specific conditions (Black Americans of ≥80 years)	aOR	Good	Reference	
				Poor	1.9 (1.1–3.2)	S
		No specific conditions (White Americans of ≥80 years)		Good	Reference	
				Poor	2.0 (1.7–2.5)	S
Van den Brink et al., 2005 [29]	10-year mortality	No specific conditions (only men born between 1900 and 1920)	aHR	Healthy	Reference	
				Not healthy	1.19 (0.97–1.46)	NS
Baron-Epel et al., 2004 [30]	91-month mortality	No specific conditions (men)	aHR	Sub-optimal	Reference	
				Optimal	1.33 (1.10–1.61)	<0.01
		No specific conditions (women)	aHR	Sub-optimal	Reference	
				Optimal	1.40 (1.17–1.67)	<0.01
Walker et al., 2004 [31]	5-year mortality	No specific conditions, cognitively intact	aHR	Good	Reference	
				Poor	1.57 (1.38–1.78)	S
		No specific conditions, mild to moderate cognitive impairment		Good	Reference	
				Poor	1.26 (1.01–1.59)	S
		No specific conditions, severe cognitive impairment		Good	Reference	
				Poor	1.00 (0.76–1.31)	NS
Bath, 2003 [32]	4-year mortality	No specific conditions (men)	aHR	Excellent	Reference	
				Good	0.67 (0.35–1.29)	NS
				Average	1.17 (0.55–2.50)	NS
				Fair	0.62 (0.25–1.53)	NS
				Poor	0.87 (0.32–2.33)	NS
		No specific conditions (women)	aHR	Excellent	Reference	
				Good	1.44 (0.63–3.29)	NS
				Average	1.15 (0.44–2.98)	NS
				Fair	1.13 (0.41–3.06)	NS
				Poor	1.98 (0.63–6.25)	NS
	12-year mortality	No specific conditions (men)	aHR	Excellent	Reference	
				Good	0.94 (0.66–1.34)	NS
				Average	1.16 (0.73–1.83)	NS
				Fair	1.01 (0.61–1.66)	NS
				Poor	1.54 (0.84–2.83)	NS
		No specific conditions (women)	aHR	Excellent	Reference	
				Good	1.09 (0.76–1.57)	NS
				Average	0.84 (0.54–1.31)	NS
				Fair	1.17 (0.75–1.84)	NS
				Poor	1.30 (0.72–2.36)	NS
Helmer et al., 1999 [33]	5-year mortality	No specific conditions	aHR	Very good	Reference	
				Good	1.93 (1.15–3.23)	<0.05
				Fair	2.01 (1.16–3.46)	<0.05
				Bad/very bad	1.87 (0.99–3.55)	NS
Yu et al., 1998 [17]	5-year mortality	No specific conditions (aged 65–74 years)	aHR	Excellent/good	Reference	
				Fair	2.16 (1.44–3.25)	<0.001
				Poor	1.93 (1.20–3.11)	0.007
		No specific conditions (aged 75 years and older)		Excellent/good	Reference	
				Fair	1.14 (0.87–1.49)	0.338
				Poor	1.34 (0.95–1.88)	0.092
Leung et al., 1997 [34]	3-year mortality	No specific conditions (living in institutions)	aHR	Good	Reference	
				Average	4.05 (0.93–17.70)	NS
				Fair/poor	6.00 (1.39–25.90)	S
Schoenfeld et al., 1994 [35]	3-year mortality	Aging successfully	aOR	Excellent	Reference	0.0001 ^#^
				Good	2.69 (2.15–3.38)	
				Fair	7.26 (4.61–11.44)	
				Poor/bad	19.56 (9.89–38.68)	
Tsuji et al., 1994 [16]	3-year all-cause mortality	No specific conditions	aHR	Excellent/good	Reference	
				Fair	2.23 (1.53–3.26)	S
				Poor	3.07 (1.50–6.26)	S
	3-year cancer mortality			Excellent/good	Reference	
				Fair	3.41 (1.86–6.24)	S
				Poor	13.61 (3.47–53.42)	S
	3-year stroke mortality			Excellent/good	Reference	
				Fair	2.44 (0.97–6.15)	NS
				Poor	2.48 (0.68–9.07)	NS
	3-year heart disease mortality			Excellent/good	Reference	
				Fair	0.96 (0.32–2.86)	NS
				Poor	1.34 (0.21–8.50)	NS
Pijls et al., 1993 [13]	5-year all-cause mortality	No specific conditions (men)	aHR	Healthy	Reference	<0.001 ^#^
				Rather healthy	1.3 (0.9–1.8)	
				Moderately healthy	2.4 (1.5–3.8)	
				Not healthy	5.4 (2.7–11.0)	
	5-year cardiovascular diseases mortality			Healthy	Reference	0.09 ^#^
				Rather healthy	1.3 (0.8–2.2)	
				Moderately healthy /not healthy	1.9 (0.9–3.8)	
	5-year cancer mortality			Healthy	Reference	0.003 ^#^
				Rather healthy	1.1 (0.6–2.1)	
				Moderately healthy /not healthy	4.2 (1.9–9.4)	
Rakowski et al., 1993 [14]	Long-term mortality	No specific conditions	OR	Excellent	Reference	
				Very good	1.22 (0.98–1.53)	NS
				Good	1.48 (1.21–1.82)	S
				Fair	2.40 (1.93–3.00)	S
				Poor	4.49 (3.50–5.77)	S

95% CI, 95% confidence interval; aHR, adjusted hazard ratio; OR, non-adjusted odds ratio; aOR, adjusted odds ratio; S, significant; NS, not significant; ^#^
*p* for trend.

**Table 3 ijerph-20-03813-t003:** Quality assessment of the different studies included in this systematic review performed using the Newcastle–Ottawa scale (NOS).

Author(s), Year	Study Design	Selection	Comparability	Outcome	Total Score	Quality Rating
Wuorela et al., 2020 [18]	Longitudinal	****	**	***	9	High
Godaert et al., 2018 [11]	Longitudinal	****	**	***	9	High
Godard-Sebillotte et al., 2016 [12]	Longitudinal	****	**	***	9	High
Mavaddat et al., 2016 [19]	Longitudinal	***	**	***	8	High
Brown et al., 2015 [20]	Longitudinal	***	**	***	8	High
Gurland et al., 2014 [21]	Longitudinal	****	**	***	9	High
Shen et al., 2014 [15]	Longitudinal	****	**	***	9	High
Fernández-Ruiz et al., 2013 [22]	Longitudinal	****	**	***	9	High
Puts et al., 2013 [2]	Longitudinal	***	**	***	8	High
Ernstein et al., 2011 [23]	Longitudinal	***	**	***	8	High
Khang et al., 2010 [24]	Longitudinal	***	**	**	7	High
Ford et al., 2008 [25]	Longitudinal	***	**	***	8	High
Johansson et al., 2008 [26]	Longitudinal	***	**	***	8	High
Okamoto et al., 2008 [27]	Longitudinal	****	**	***	9	High
Lee et al., 2007 [28]	Longitudinal	****	**	***	9	High
Van den Brink et al., 2005 [29]	Longitudinal	***	**	***	8	High
Baron-Epel et al., 2004 [30]	Longitudinal	****	**	***	9	High
Walker et al., 2004 [31]	Longitudinal	****	**	***	9	High
Bath, 2003 [32]	Longitudinal	****	**	***	8	High
Helmer et al., 1999 [33]	Longitudinal	****	**	***	9	High
Yu et al., 1998 [17]	Longitudinal	****	**	***	9	High
Leung et al., 1997 [34]	Longitudinal	**	**	***	7	High
Schoenfeld et al., 1994 [35]	Longitudinal	****	**	***	9	High
Tsuji et al., 1994 [16]	Longitudinal	***	**	***	8	High
Pijls et al., 1993 [13]	Longitudinal	***	**	***	8	High
Rakowski et al., 1993 [14]	Longitudinal	****	**	***	9	High

NOS scores ≥7 were considered to indicate high-quality studies, and scores of 5–6 indicated moderate quality. The sum of the stars constitutes the Total score (for the first row: 4 stars for selection, two stars for Comparability, and three stars for outcome equal 9 stars (total score equals 9).

## Data Availability

Data can be make available at moustapha.drame@chu-martinique.fr.

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
