# Peer review of "Self-Rated Health as a Predictor of Mortality in Older Adults: A Systematic Review"

_ijerph, 2023, doi:10.3390/ijerph20053813_

Round 1
Reviewer 1 Report
I think this is a fine paper that warrants publication. My only quibble is minor. I would like to see a bit of additional attention to the (future) role of monitoring devices (e.g., FitBit, Apple watch, etc.) that are increasingly popular and providing a potential for medically relevant information and longitudinal data for analysis.
On a editorial point, please break up the overly long paragraph in the discussion into multiple paragraphs.
Author Response
Please see file attached

Reviewer 2 Report
Thank you for inviting me to review this work. This systematic review entitled "Self-Rated Health as a Predictor of Mortality in Older Adults: A 2 Systematic Review", aimed to investigate the link between self-reported health (SRH) and mortality in older adults.
Minor comments have been raised:
Introduction:
-What do you mean by older adults? According to WHO, Please define them and add the age cutoff point of this group.
Methods:
This sentence is unclear "The intervention (exposure) and the comparator were the levels of self-rated health (SRH)."
Please rephrase this sentence: " selection, comparability, and outcome assessment, assigning a maximum of four 92 points for selection, two points for comparability, and three points for outcome."
Results: This section is very well presented.
Figure 1 appears blurry, please re-insert it
Discussion:
I believe this section needs elaboration and more deep interpretations for the results.
Please clarify what you mean by "The majority of articles concerned populations not selected for the presence of a specific pathology at inclusion"
Please elaborate more after this sentence "Zajacova et al [47] point out that all health indicators are significantly 39 associated with SRH" What are the discussed health indicators?
Conclusion:
I believe this is a strong sentence to end with, "SRH is a good criterion for assessing the risk of mortality in the short, medium or long term in a population of elderly subjects living at home."; hence, the discussion did not clarify. Please rephrase it.
Reviewer 3 Report
This manuscript investigated the correlation between self-reported health status and mortality in older adults. This is a really interesting and promising study. The study based on the analysis of several open source data bases. The manuscript was written with using high scientific level, appropriate methods. The results are clear.
I attach a number of considerations that I consider to be of interest and should be improved:
1.L.27 - Avoid mass-citation
2. Requires more detailed analysis (L.116, 118,132)
3. L.72 - Emphasize most successful application. It's not clear enough how did you make this conclusion.
4.Table 1, Column "Study setting, Medical condition" - Please describe the principles of sorting the order
5.Table 2. Please describe how did you choose the metrics Optimal/ Good /...etc. You could show a numerical scale of evaluation
6. Could you present your results in graphs for more clarity?
Round 2
Reviewer 3 Report
Dear Authors,
Thanks for the detailed answers to the questions. Despite all changes and additions you’ve made, I still believe that the principles of sorting the order could be more appropriate. Moreover, the results in present form are not clearly presented and hard to read.
Author Response
x
